# Watching the release of a photopharmacological drug from tubulin using time-resolved serial crystallography

Maximilian Wranik[1], Tobias Weinert [1], Chavdar Slavov [2], Tiziana Masini[3], Antonia Furrer [1], Natacha Gaillard[1], Dario Gioia[3], Marco Ferrarotti[3], Daniel James[1], Hannah Glover [1], Melissa Carrillo [1], Demet Kekilli[1], Robin Stipp[1], Petr Skopintsev[1], Steffen Brünle[1], Tobias Mühlethaler[1], John Beale [1], Dardan Gashi[4], Karol Nass[4], Dmitry Ozerov [5], Philip J. M. Johnson [4], Claudio Cirelli [4], Camila Bacellar [4], Markus Braun [2], Meitian Wang [4], Florian Dworkowski [4], Chris Milne [4], Andrea Cavalli [3,6], Josef Wachtveitl [2], Michel O. Steinmetz [1,7] ✉ & Jörg Standfuss [1] ✉

The binding and release of ligands from their protein targets is central to fundamental biological processes as well as to drug discovery. Photopharmacology introduces chemical triggers that allow the changing of ligand affinities and thus biological activity by light. Insight into the molecular mechanisms of photopharmacology is largely missing because the relevant transitions during the light-triggered reaction cannot be resolved by conventional structural biology. Using time-resolved serial crystallography at a synchrotron and X-ray free-electron laser, we capture the release of the anti-cancer compound azo-combretastatin A4 and the resulting conformational changes in tubulin. Nine structural snapshots from 1 ns to 100 ms complemented by simulations show how *cis*-to-*trans* isomerization of the azobenzene bond leads to a switch in ligand affinity, opening of an exit channel, and collapse of the binding pocket upon ligand release. The resulting global backbone rearrangements are related to the action mechanism of microtubule-destabilizing drugs.

The dynamic interaction of small molecule ligands with their target proteins is the basis for numerous physiological processes in living cells. To catalyze chemical reactions, enzymes have to bind their substrates and release the products. Cellular signaling is coordinated by the binding of ligands that activate their cognate receptors in the cytoplasm or the cellular membrane. Small molecule ligands that act as drugs to selectively intercede in these physiological processes provide the most common strategy for medical intervention.

Photopharmacology introduces photochemical triggers into small molecule ligands to be able to change binding affinities and manipulate complex biological processes with the help of light[1]. The most widely used photochemical affinity switches are based on the

[1]Division of Biology and Chemistry, Paul Scherrer Institut, 5232 Villigen, Switzerland. [2]Institute of Physical and Theoretical Chemistry, Goethe University, Frankfurt am Main, Germany. [3]Computational & Chemical Biology, Istituto Italiano di Tecnologia, 16163 Genova, Italy. [4]Photon Science Division, Paul Scherrer Institut, 5232 Villigen, Switzerland. [5]Scientific Computing, Theory and Data, Paul Scherrer Institut, 5232 Villigen, Switzerland. [6]Department of Pharmacy and Biotechnology, University of Bologna, 40126 Bologna, Italy. [7]Biozentrum, University of Basel, 4056 Basel, Switzerland. ✉e-mail: michel.steinmetz@psi.ch; joerg.standfuss@psi.ch

azobenzene scaffold that allows for high quantum efficiencies, fast trigger rates, and reversible photoisomerization upon illumination[2]. A prominent example is an azobenzene derivative of combretastatin A4 (azo-CA4)[3], a photopharmacological compound that binds the colchicine site of the αβ-tubulin heterodimer (hereafter called tubulin)[4]. Tubulin is the key building block of the microtubule cytoskeleton in all eukaryotic cells and among others essential for cell division[5]. It is thus a common target for medically relevant agents including the first-line anti-cancer drug paclitaxel (Taxol). While colchicine modulates multiple pro- and anti-inflammatory pathways associated with gouty arthritis[6], combretastatin A4 (CA4) is a strong cell growth inhibitor[7] that has been tested in clinical trials against different types of tumors[8,9]. Interestingly, the phase III drug sabizabulin, which decreases mortality in intense-care COVID-19 patients by 55%[10], is also a colchicine-site ligand, sharing one binding moiety with combretastatins. The diverse applications of these small molecule drugs highlight the importance of the colchicine site as a hotspot to influence the various cellular processes related to microtubule structure and function. A better understanding of how the dynamic interplay between a ligand and the colchicine site is related to the conformational plasticity of tubulin may help further developments of this class of anti-tubulin agents for medical applications.

Besides its applications in photopharmacology[1] and material sciences[11], the azobenzene scaffold is also ideally suited for time-resolved pump–probe spectroscopy to understand protein motion[12]. Among the latest incarnations of the pump–probe technique is time-resolved serial crystallography to resolve molecular snapshots of proteins that can be assembled into stop-motion movies of how proteins function[13,14]. The method has been most successful when studying photoactive proteins that are easy to trigger with high temporal precision; however, these represent less than 0.5% of all known proteins[15]. In order to reach the full potential of time-resolved crystallography, new ways have to be developed to trigger protein–ligand interaction and protein dynamics in non-photoactive protein targets.

To this end, we show here how to utilize photopharmacological compounds to assess protein–ligand interaction dynamics and the resulting protein response over time. We first obtained the structure of azo-CA4 in complex with tubulin, before following its photoflash-induced release at the Swiss Light Source (SLS) and the Swiss X-ray Free Electron Laser (SwissFEL). We resolved the structural evolution at up to 1.7 Å spatial resolution in nine logarithmically timed structural snapshots ranging from 1 ns to 100 ms. Our molecular snapshots show the formation and dissolution of a metastable, intermediate ligand-binding pose, and the accompanying reorganizations of the most frequently targeted drug-binding site in tubulin[16]. We anticipate that similar photoswitchable compounds developed to control kinases, channels, G protein-coupled receptors, and other pharmaceutically relevant protein targets will be important future tools to study protein–ligand interaction dynamics in non-photoactive protein targets at high spatial resolution and without compromising time resolution.

## Results
### Adapting photopharmacology for time-resolved serial crystallography
Time-resolved crystallography is ever more sample efficient (Supplementary Fig. 1) and the technology is now available at a growing number of synchrotrons and X-ray free-electron lasers (XFELs)[17,18]. With the increased availability of beamtime at these advanced X-ray sources, future challenges will shift toward enabling dynamic studies on a wide range of relevant biological targets. Photopharmacology could be an important part of the solution as the field already developed a variety of photosensitive compounds that can be utilized as efficient triggers to activate diverse soluble and membrane proteins of medical interest (for some examples, see Supplementary Fig. 2).

In the last few years, tubulin-targeting photoswitches based on different scaffolds have been synthesized, for some of which we also obtained structures[19,20]. To demonstrate the use of a photochemical affinity switch for time-resolved crystallography, we chose azo-CA4 as a prominent photopharmacological compound[21] with minimal modifications to the parent compound CA4. Replacing the central C=C stilbene bond found in CA4 with an N=N azobenzene bond in azo-CA4 results in a shift of the absorption maximum out of the ultraviolet range and in an increased photoreactivity. After the modification, illumination with blue or green light allows for reversible switching of the molecule between the high-affinity *cis* and low-affinity *trans* conformations, respectively[3]. This effect allows perturbing the division of cancer cells using a localized pulse of monochromatic light[3]. However, how azobenzene isomerization changes binding affinity, how this induces structural adaptations within the binding pocket, and how this then affects the global conformational plasticity of tubulin that is needed during microtubule formation is largely unknown.

The idea behind our experiment was to use time-resolved crystallography to follow how the ligand relaxes after photoactivation, allowing it to be observed bound to the protein in its out-of-equilibrium, low-affinity state, and eventually observe ligand unbinding and the resulting changes in the protein (Fig. 1). As a first step, we synthesized azo-CA4 and characterized its tubulin-binding properties under crystallization conditions using the TD1 crystal system composed of one αβ-tubulin heterodimer and one DARPin D1 molecule[22,23] (Supplementary Fig. 3). Based on these results, we designed time-resolved crystallography experiments to reveal the structural changes upon ligand isomerization and release.

Overall, we collected 1'139'945 indexable diffraction patterns with a resolution of up to 2.1 Å when using serial synchrotron crystallography at the SLS[24,25] and 1.7 Å when using serial femtosecond crystallography at the SwissFEL[26] (Supplementary Table 1). The XFEL data are divided into "dark" images collected without light exposure, images from crystals without ligand, and pump–probe data taken at logarithmically spaced time delays of 1 ns, 10 ns, 100 ns, 1 µs, 10 µs, 100 µs, 1 ms, and 10 ms after illumination with flashes of laser light.

This untargeted approach relies on both synchrotron and XFEL setups to cover nine orders of magnitude in time, which increases accessibility and remains an option even for cases where initial spectroscopic data is scarce. We have characterized the ultrafast regime and initial photochemical reaction using transient absorption spectroscopy (Supplementary Fig. 3); however, here we focus on the temporal domain starting from the nanoseconds where the biochemically most relevant conformational changes in the protein are to be expected[18].

Difference electron density maps ($F_{obs}^{light} - F_{obs}^{dark}$) calculated from the time-resolved data are of high quality (Fig. 1D, Supplementary Fig. 4, Supplementary Movie 1) and can be qualitatively followed by Pearson correlation analysis to highlight transitions between structural intermediates[27] (Fig. 1B). We observed three major blocks of elevated correlation compatible with a nonequilibrium photoproduct population that relaxes via several intermediates into the ligand-free state of tubulin. Following the build-up of negative density around azo-CA4 (Fig. 1C) suggests that ligand release starts after the 1 ms delay at the end of the second block observed in the Pearson correlation analysis. Using structural refinements against extrapolated data from the light-activated fraction of the structure factors[28,29], we were able to generate atomic models for predominant intermediate states (Fig. 1D, Supplementary Table 2). In addition, we used conventional refinement approaches to obtain room-temperature crystal structures of tubulin in its *cis*-azo-CA4 bound and ligand-free states.

Together, these data provide a comprehensive overview of the structural changes occurring after *cis-trans* isomerization and the following release of the azo-CA4 compound from its tubulin binding pocket. A molecular movie obtained from morphing between these

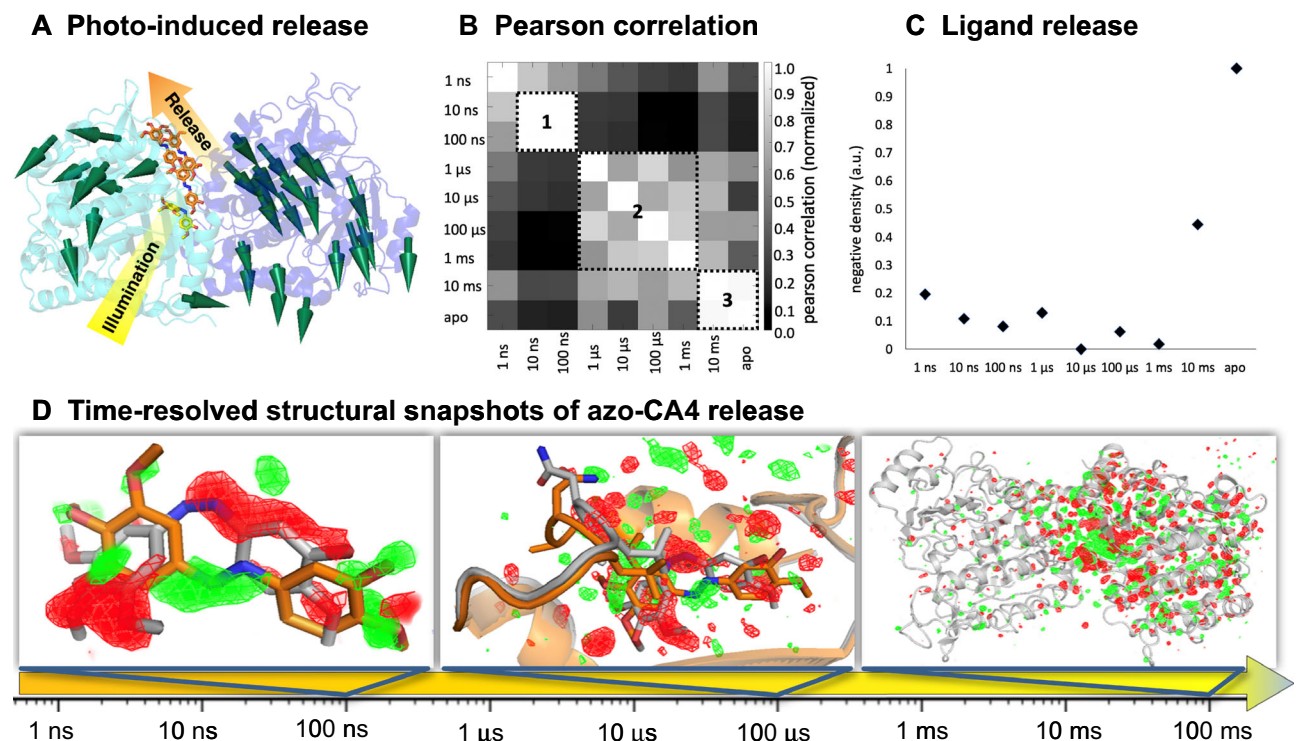

**Fig. 1 | Tracking ligand release from tubulin using a photochemical affinity switch. A** The cis-azo-CA4 (yellow sticks) binds to the colchicine site that is formed by secondary structural elements stemming from both the α (blue) and β (cyan) subunits of the αβ-tubulin heterodimer. Illumination with a flash of a laser light triggers the release of *trans*-azo-CA4 (colored in orange, diffusion indicated by including several molecules, cylindrical arrows designate conformational changes of the protein) from its binding pocket. **B** Difference electron density maps ($F_{obs}^{light} − F_{obs}^{dark}$) from time-resolved XFEL data were integrated and subjected to Pearson correlation analysis. Three areas of elevated correlation can be identified. The box in the nanoseconds (1) corresponds to the relaxed conformation of the ligand observed as a metastable binding pose. Correlations between maps in the microseconds (2) are less homogeneous and stretch from 1 μs to the opening of the

exit pathway at 1 ms. The final snapshots in the milliseconds (3) correspond to the predominantly unliganded protein. **C** Plot showing negative density integrated around azo-CA4 with a strong increase after 1 ms indicating release of the ligand. **D** Time-resolved structural snapshots of azo-CA4 release. The time arrow depicts the investigated time regime. The panels from left to right show the isomorphous difference maps obtained at 100 ns with changes centered on the ligand, 100 μs with changes centered on the binding pocket, and 100 ms with conformational changes propagating throughout the protein. All panels show isomorphous difference maps in red (negative) and green (positive) at 3 σ. The structure in the given time range (colored in orange) is compared to that of the previous time range (colored in gray). A movie showing the electron changes within the binding pocket over all nine time delays is available as Supplementary material.

states (Supplementary Movie 2) summarizes the critical molecular rearrangements that are described in the following.

## Formation of a metastable binding pose after azo-CA4 photoisomerization

The conformation of a ligand is generally presumed to precisely fit the architecture of its binding pocket. The photoinduced release of azo-CA4 provides a system to study what happens when a ligand deviates from its primary binding pose (Fig. 2A). Before light activation, the *cis* isomer of azo-CA4 is bound to the colchicine site at the interface between the α and β subunits of tubulin (Fig. 1A). Interactions with the protein are primarily hydrophobic in nature, except for hydrogen bonds to the backbone carbonyl of αThr179 and the backbone amide of αVal181, respectively (Fig. 2B). Further interactions are mediated by a water molecule interacting directly with the azobenzene bond of azo-CA4. Overall, the binding pose of the ligand is nearly identical to that of the parent compound CA4 which contains a stilbene instead of the azobenzene group connecting the A and B rings of the ligand[4].

Transient absorption spectroscopy shows how the illumination of azo-CA4 induces *cis-trans* isomerization around the azobenzene bond, which is completed in the late picosecond range followed by its relaxation into the nanoseconds (Supplementary Fig. 3). Our structures show how the compound transitions from the partially disordered conformation observed in the 1 ns time delay into a relaxed *trans*-azo-CA4 state observed at 100 ns after light activation. At this point in time, the molecule is stretched by 2.2 Å, which relocates its A

ring by 2.6 Å toward the βT7 loop known to be involved in ligand unbinding[4,23] (Fig. 2C). Stretching along the azobenzene bond changes the interaction pattern with the protein and the binding pocket. The water-mediated interactions are broken and relocation of the two ring moieties of azo-CA4 reduces their hydrophobic stacking interactions with the protein.

To bridge the gap between structural effects and binding energy we relied on molecular dynamics (MD) simulations[30]. We aimed at estimating the relative binding free energy difference between the *cis* and *trans* isomers ($\Delta\Delta G_{cis\text{-}trans}$) using the computational protocol utilized for the tubulin-combretastatin A4 complex[4]. As expected, the *trans* isomer is the most stable form in both the bound and unbound simulations with ΔG values of $\Delta G_{cis\text{-}trans\text{-}bound} = −14$ kcal/mol and $\Delta G_{cis\text{-}trans\text{-}unbound} = −19$ kcal/mol, respectively. The resulting relative binding free energy difference of $\Delta\Delta G_{cis\text{-}trans} = 5$ kcal/mol clearly favors the *cis*-isomer stabilized interactions with tubulin (free energy profiles are given in Supplementary Fig. 5). Such a free energy difference translates into an increased affinity of ~4300 times based on the equilibrium probability ratio and Boltzmann statistics. This change is in good agreement with radioligand scintillation proximity assays that indicated an $EC_{50}$ value of ~30 μM for illuminated azo-CA4 (representing a mixture of *cis-trans* isomers) but no significant competitive binding of the relaxed *trans*-azo-CA4 to tubulin[3]. The low affinity of *trans*-CA4 also explains why we were unable to crystallize this isomer in complex with tubulin using classical crystallography

**Fig. 2 | Photoisomerization leads to the formation of a metastable binding state. A** The two chemical structures of azo-CA4 show the wavelength-dependent conformational change between the *cis* (top) and *trans* (bottom) conformations. Central to the switch is the N=N azo bond that allows to switch azo-CA4 reversibly and with high efficiency between its high- and low-affinity stereoisomers, respectively. **B** Before laser-induced isomerization, *cis*-azo-CA4 (yellow sticks) is bound in the colchicine site located between the α (blue) and β (cyan) subunits close to the βT7 loop (green) of β-tubulin. Hydrophobic interactions (green lines) between tubulin residues and the A and B rings of *cis*-azo-CA4 as well as hydrophilic interactions (blue dashed lines) toward the ring substituents anchor the ligand within the colchicine site. **C** Within nanoseconds after illumination, *trans*-azo-CA4 (orange sticks) has relaxed into a metastable binding pose with an altered interaction network and reduced affinity. For clarity, only interacting residues are shown.

methods but instead needed a time-resolved experiment to resolve the metastable binding pose before azo-CA4 release.

**Binding pocket changes upon azo-CA4 release**

The colchicine site undergoes pronounced conformational changes between the ligand-bound and -unbound states that are incompatible with a simple "lock-and-key" model of protein–ligand interactions[4,31–33]. In our time-resolved data, the relevant temporal regime starts at 100 ns, when the A ring of the ligand has been repositioned into close contacts with residue βAla248 of the βT7 loop (Fig. 3A). Initial conformational changes of the βT7 loop follow on the microsecond scale with further rearrangements of the loop, including a displacement of βAsn247 and βLeu246 occurring in the millisecond time range. The binding pocket initially expands starting from a volume of 446 Å³ before illumination (*cis*-azo-CA4) to 525 Å³ at 100 ns (relaxed *trans*-azo-CA4) and then collapses from 363 Å³ at 1 ms (after βT7 loop movement) down to 71 Å³ at 10 ms when the compound has left the binding pocket. Based on visual inspection of difference maps and plotting electron density changes on the ligand over time, we have modeled the release of azo-CA4 between the 1 and 10 ms time delays (Fig. 1C). However, the residual ligand can still be observed at low occupancy in the 10 ms data and is only fully released at about 100 ms, where the structure is essentially identical to the one determined in the absence of azo-CA4 (Supplementary Fig. 6). The long release time is consistent with a stochastic nature of the process

where unbinding events of individual molecules occur over several milliseconds in time once they are initiated on faster timescales.

The binding pocket is not freely accessible to the solvent but our structural analysis suggests the opening of an exit channel between 100 μs and 1 ms (Fig. 3B). Reorganization of the βT7 loop, particularly through a rotation of the βLeu246 side chain, results into an 8-Å-long channel forming a possible exit site close to the A ring of the ligand (Fig. 3C). The diameter of the channel is 4–6 Å, which seems small given the 7 Å diameter for the larger A ring of azo-CA4. As the unbinding takes place over many milliseconds throughout the crystal, it seems likely that the channel transiently widens in short-lived intermediates that did not sufficiently accumulate to be observed. To study this possibility, we have used MD simulations (Fig. 3D) starting from the 1 ms delay where the opening of the channel is most prominent. Almost 50 μs of simulations (Supplementary Data 1 and 2) suggest that the azo-CA4 approaches a position close to the proposed exit site of the protein, where only minimal changes would be sufficient to let the ligand diffuse out of its binding pocket. Once the ligand detaches, our crystallographic data again shows how the βT7 loop flips back and packs against β-tubulin residues that were previously engaged in azo-CA4 binding.

Together, these data are consistent with the flexible βT7 loop to act as a "molecular gate" that opens and closes the colchicine site and, depending on its current conformation, directly

## A Binding pocket adaptations over time

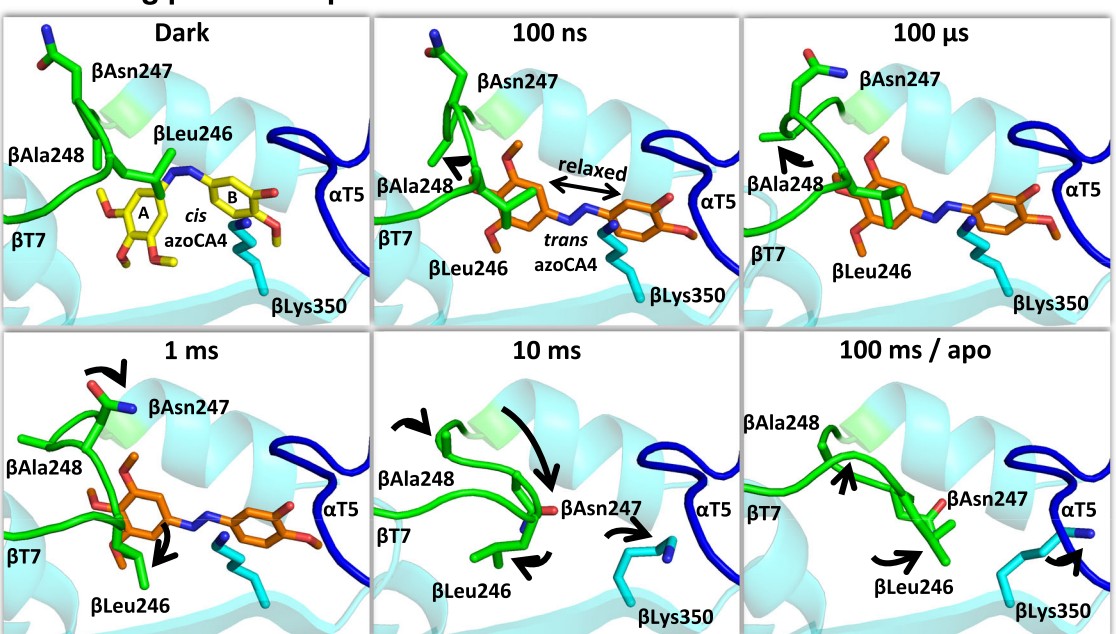

## B Molecular gating of ligand release

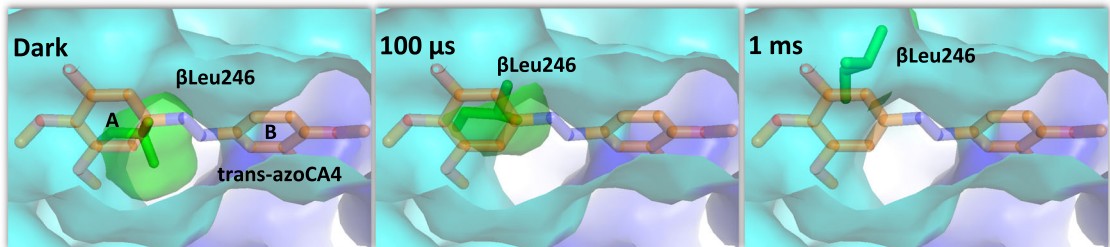

## C Pathway identification    D Molecular dynamics simulation

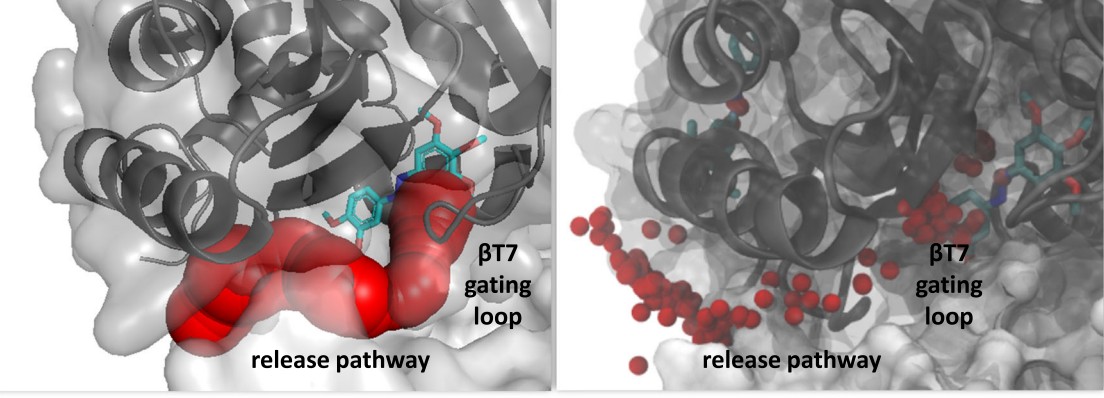

**Fig. 3 | Molecular changes in the βT7 gating loop allow ligand release. A** Within nanoseconds, the light-induced isomerization and relaxation repositions the A ring of azo-CA4 closer to the βT7 loop (green), which acts like a lid on the colchicine site. Further reorganizations (black arrows) in the microseconds open a channel between the βH8 helix and the βT7 loop. In the millisecond range, the βT7 loop is folded back and packed against residues of the empty binding pocket. **B** Opening of a release pathway through relocation of the βT7 loop in the 1 ms structure is illustrated by the position of βLeu246 (green sticks and surface) at the indicated time delays. **C** The release pathway (red spheres) plotted onto the 1 ms structure. **D** The experimentally deduced unbinding pathway of azo-CA4 is confirmed by MD simulations. The small red spheres depict the centers of masses of ligand atoms, plotted every 50 frames along the simulation.

competes with ligand binding[32,33]. Notably, all colchicine-site ligands must displace the βT7 loop upon binding[16,34] as the pocket is contracted fully in ligand-free tubulin structures. Along with the dynamic nature of the pocket at room temperature[24], this observation suggests conformational selection as the predominant mechanism of how microtubule-destabilizing agents interact with the colchicine site.

## Global backbone rearrangements

Tubulin displays substantial structural plasticity that is key to microtubule dynamics and thus its cellular function. This includes the global conformational changes from a "curved" free tubulin to a "straight" microtubule-lattice incorporated tubulin conformation[35]. This reorganization of the tubulin dimer is accompanied by a contraction of the colchicine site[32,33]. Ligand binding prevents pocket contraction and

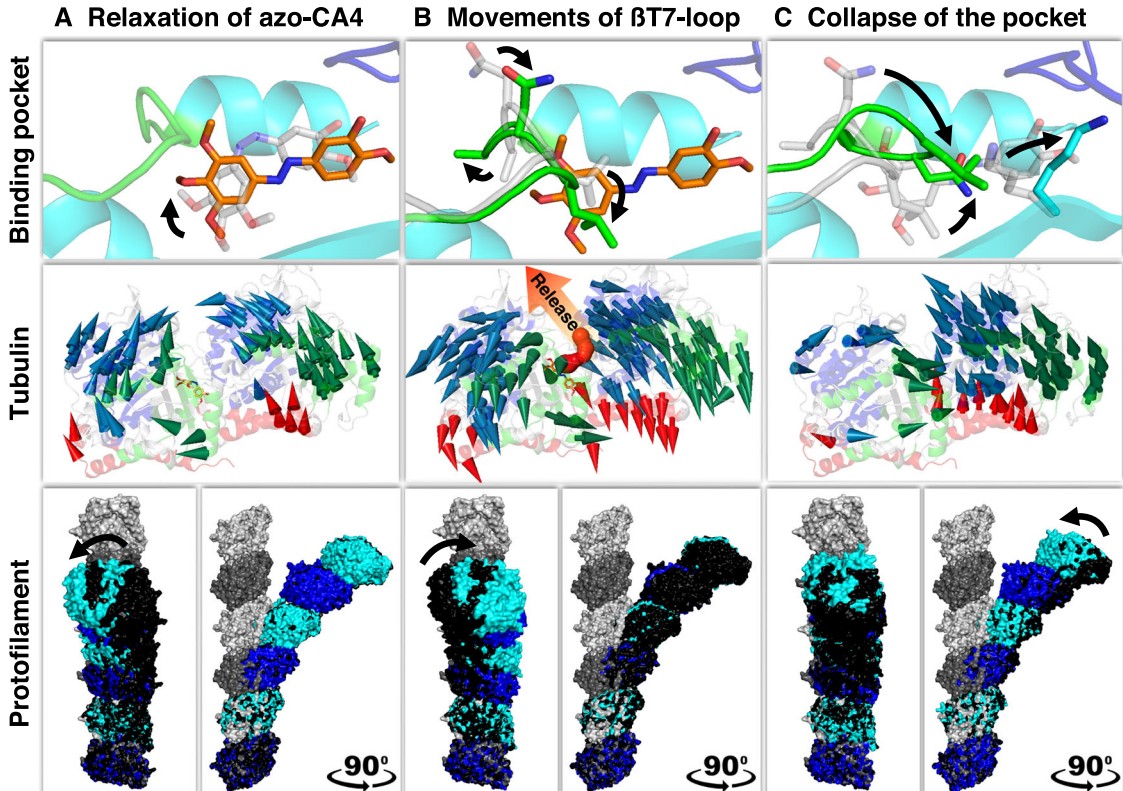

**Fig. 4 | Correlation between colchicine-site dynamics and global tubulin backbone rearrangements. A** Relaxation of azo-CA4: dark to 100 ns. **B** Movements of βT7 loop: 100 ns to 1 ms. **C** Collapse of the colchicine site: 1 ms to 100 ms. The three rows depict conformational changes at different spatial levels: Binding pocket in the upper row, tubulin backbone in the middle row, and longitudinally aligned tubulin dimers in the lower row. Starting structures are shown in gray. Backbone movements within tubulin (middle row) are represented by modevectors between consecutive time delays. Every second Cα-backbone atom in the main secondary structural elements with a displacement of at least 0.5 Å is shown. The N-terminal nucleotide-binding domains, the C-terminal domains, and the intermediate domains for both α and β tubulin are shown in blue, red, and green, respectively. The release pathway (red spheres) is indicated to illustrate the directional movements of the tubulin subunits. The global conformational changes (lower row) are visualized by three longitudinally aligned tubulin dimers ("Straight" tubulin dimers as found in the main shaft of a microtubule are shown in surface representation (dark gray, α-tubulin; light gray, β-tubulin; PDB ID 5SYE); "Curved" tubulin dimers (α-tubulin in blue and β-tubulin in cyan) at the indicated time delay and in relation to the previous state are shown in black). Deviations demonstrate the conformational plasticity of tubulin during azo-CA4 relaxation in its binding pocket and βT7 loop movements of β-tubulin, as well as a directed curvature adjustment after azo-CA4 release. Arrows indicate the main directions of movements in the indicated time delays.

thus stabilizes the curved conformation[16] as the principal action mechanism for many tubulin-targeting anti-cancer drugs[16,32,33,35].

To illustrate how global backbone rearrangements develop during the release of azo-CA4, we superimposed our time-resolved structures and visualized the rearrangements of Cα backbone atoms as individual vectors. We observed several motions that correspond well with our initial Pearson correlation analysis of electron density changes, and which coincide in time with the major events in the ligand-binding pocket (Fig. 4). Global structural changes are already present in the nanosecond time range, where we observed coordinated inward motions of both the α- and β-tubulin subunits (Fig. 4A). These changes correlated well to the time range where the A ring of azo-CA4 shifts to a new place completing the *cis*-to-*trans* relaxation process within the binding pocket. Interestingly, some of these movements are reverted in the microsecond time range, where the βT7 loop starts moving (Fig. 4B). At this stage of the reaction, global movements in the α and β subunits are directed predominantly away from the colchicine site and may cause opening of a ligand release pathway at 1 ms. In the later millisecond range, where the βT7 loop finally collapses into the binding pocket, backbone movements are primarily located in the α-tubulin subunit and bring the structure back to its ligand-free state (Fig. 4C).

To analyze whether the global backbone rearrangements could have an impact on the overall conformation of microtubules, we analyzed the curvature of an oligomer constructed from three longitudinally, head-to-tail aligned tubulin dimers, as previously described in ref. [36]. Indeed, the curvature is reduced significantly toward the direction of the straight protofilament conformation found in microtubules once azo-CA4 is released. Our time-resolved structures thus suggest that the global tubulin backbone movements, as well as overall changes in tubulin conformational curvature, evolve on par with structural alterations of the ligand that is bound to the colchicine site.

## Discussion

In the field of photopharmacology ligand design is done largely "blind" in the sense that little is known about the molecular mechanism of how these modifications will alter the dynamics and activity of the target protein. In this context, it was reassuring to see that the interaction pattern of the parent compound CA4 and the photoactive azo-CA4 remains largely the same. Further knowledge of both the initial binding mode and the metastable binding pose after illumination provides the molecular basis to understand differences in binding affinity between the *cis* and *trans* forms of the compound. Together with molecular information on how the pocket adapts during ligand release, this may provide innovative angles to improve the potency of azo-CA4 and perhaps other photopharmaceutical compounds.

In the wider context of general protein–ligand interaction dynamics, our work provides molecular information beyond the

classical "key and lock" model. The pharmaceutical industry relies on such information as an integral part of a rational approach to discover new drugs and understand their mechanisms of action. The colchicine site is the most frequently targeted drug-binding site in tubulin[16] with broad relevance to combat gout[6], cancer[7–9], and COVID-19[10]. Following the relaxation and release of the same ligand within different isoforms of tubulin[23] or characterizing ligands specific for tubulin variants of parasitic origin[37] may offer new avenues to rationally design innovative anti-tubulin ligands.

Our study also demonstrates how compounds developed for photopharmacology can be used for time-resolved crystallography. While promising results using photo-caged compounds[38] and rapid mixing have been shown[39,40], these approaches will not allow observations on the fast timescales relevant for protein–ligand interactions. It is furthermore difficult to predict how ligands diffuse within a crystal, as this depends on crystal shape, size, packing, and solvent content among other parameters[41]. Starting a time-resolved experiment with a prebound ligand is an elegant solution to the problem, even though it would be interesting to study to what extent ligand binding and unbinding follow the same principles.

The underlying X-ray technology is now widely available, with many new endstations at fourth-generation synchrotrons and XFELs purpose build for time-resolved pump–probe measurements[17,18]. The rapidly expanding field of photopharmacology delivers growing numbers of photoswitches to, for example, modulate the activities of TRP channels, kinases, and G protein-coupled receptors, or to tackle antibiotic resistance[1,11,12,42–44]. Together these developments promise more direct experimental insights into the dynamic nature of protein–ligand interactions in a wide range of relevant protein targets.

## Methods

### Preparation of tubulin-DARPin D1 complex
Bovine brain tubulin was purchased from the Centro de Investigaciones Biológicas (Microtubule Stabilizing Agents Group), CSIC, Madrid, Spain. DARPin D1 was expressed and purified according to previous descriptions in ref. [22]. Detailed protocols for obtaining high-resolution tubulin crystals can be found in ref. [45].

Briefly, the pET-derived vector PSTCm9 includes the gene for the artificial DARPin D1 as an N-terminal fusion with thioredoxin followed by a hexahistidine-tag and a thrombin cleavage site. For expression, the DARPin D1-carrying plasmid was transfected into competent *E. coli* BL21 (DE3) cells, and the protein was expressed in LB medium in a 2-l Erlenmeyer flask. The cells were harvested by centrifugation at 5000×g and stored at 80 °C.

The cell walls were lysed on ice using an ultrasonic cell disrupter applying alterations of 2 s on and off pulses with 30% amplitude for 22 min. The lysate was centrifuged at 20,000×g at 4 °C for 40 min, and the supernatant was collected. The filtered supernatant was applied to a Ni-NTA column on a fast protein liquid chromatography system (FPLC). Eluted protein fractions were analyzed by SDS-Page, pooled together, and the hexahistidine-tag was cleaved using thrombin. After cleavage, the solution was applied to a Ni-NTA column on a fast protein liquid chromatography system, and the flow through collected. Suitable fractions were identified by characteristic UV absorbance at 280 nm, concentrated, and applied on a Superdex 75 16/60 column for size-exclusion chromatography. Respective fractions were pooled together, concentrated to a final DARPin D1 concentration of ~25 mg/ml using a 3-kDa cutoff concentrator, flash-frozen in liquid nitrogen in 50-µl aliquots, and stored at −80 °C. The concentration was determined in mg/ml using a Nanodrop-1000 instrument against the size-exclusion buffer using an extinction coefficient of $0.416 \, \mathrm{L g^{-1} \, cm^{-1}}$.

To extract the active and non-aggregated tubulin from the lyophilized tubulin stocks, a sequence of solution-based tubulin polymerization and depolymerization steps was performed. On ice, the lyophilized tubulin is dissolved and diluted to 10 mg/ml with a polymerization buffer containing 50 mM 2-(*N*-morpholino)ethanesulfonic acid (MES-KOH) pH 6.8, 0.5 mM ethylene glycol-bis(β-aminoethyl ether)-N,N,N′,N′-tetraacetic acid (EGTA), 0.4 mM Guanosine-5′-triphosphate (GTP), 6 mM MgCl₂, and 33% glycerol. The lyophilized pellet was dissolved over ~30 min by regular shaking and pipette resuspension on ice. Afterward, the clear solution was ultracentrifuged at 150,000 × g at 4 °C for 10 min. The supernatant was carefully extracted, transferred to pre-warmed tubes, and incubated for 25 min in a water-filled heating block at 37 °C. This procedure induces tubulin polymerization causing the solution to turn viscous and turbid. The solution was then ultracentrifuged at 300,000 × g at 30 °C for 15 min. The supernatant was carefully discarded, and the tubulin pellet was poured over with depolymerization buffer containing 80 mM Pipes-KOH pH 6.8, 0.5 mM EGTA, 2 mM Guanosine-5′-diphosphate (GDP), 1 mM MgCl₂, and 2 mM CaCl₂. After 15 min of incubation, the pellet was resuspended with a pipette, and the solution was incubated again on ice for an additional 15 min. This step was repeated until the pellet was dissolved entirely. The depolymerized tubulin solution was diluted with depolymerization buffer to a concentration of 9.8 mg/ml and stored at −80 °C.

### Tubulin crystallization
For crystallization, the tubulin-DARPin D1 (TD1) complex was formed by mixing the respective components in a 1:1.1 molar ratio. The TD1 complex was crystallized using EasyXtal 15-well plates by the hanging drop vapor diffusion method (drop size 2 µl, drop ratio 1:1, 8 drops per well) at a concentration of $9.8 \, \mathrm{mg \, ml^{-1}}$ at 20 °C with a precipitant solution containing 21% (w/v) polyethylene glycol (PEG) 3000, 0.2 M ammonium sulfate, and 0.1 M bis-tris methane, pH 5.5. All drops were subsequently hair-seeded with crystalline material obtained in previous crystallization trials to increase the homogeneity and density of crystals. After 48 h, crystals were washed off the plates with precipitant solution, collected in 0.6 ml tubes, and vortexed for a few seconds. This procedure induces batch crystallization within the tubes and a sedimented crystal pellet was formed after 24 h of incubation at 20 °C. Initially, crystals grew as long needles that were broken into smaller fragments of approximately $20 \times 20 \times 5 \, \mu\mathrm{m}^3$ during the sample preparation described below.

### Azo-CA4 synthesis
The synthetic photoswitch azo-CA4 was prepared following the experimental procedure described in ref. [3]. First, catechol (4.44 g, 40.3 mmol) was reacted with tosylchloride (9.53 g, 50 mmol) in pyridine (10 ml) for 2 h at room temperature. The reaction was quenched with aq HCl, and the organic phases extracted with dichloromethane, dried on MgSO₄, filtered, concentrated, and purified by column chromatography (gradient of cyclohexane/EtOAc, with EtOAc from 10 to 50%), resulting in the mono-tosylated product (7.5 g, 28.4 mmol) in 70% yield. Mono-tosylcatechol was then reacted with commercial 3,4,5-trimethoxyaniline via a diazo coupling with isopentyl nitrite. The aniline (297 mg, 1.62 mmol) was dissolved in methanol (8.1 ml) and concentrated HCl (0.4 ml), and the mixture was cooled in an ice bath. A solution of isopentyl nitrite (0.22 ml, 1.65 mmol) in methanol (0.97 ml) was added dropwise and the reaction was stirred for 30 min in the cold. A cold solution of the mono-tosylcathecol (450 mg, 1.7 mmol) in methanol (3.2 ml) and NaOH (2.0 M, 2.9 ml) was prepared, and to it was added the solution of the diazonium dropwise over 1 min. After 5 h stirring in the cold, the pH was adjusted to 7 with phosphate buffer, dichloromethane was added, and the aqueous phase was extracted with dichloromethane. The combined organic layers were washed with water and brine, dried on MgSO₄, filtered, concentrated, and purified by column chromatography (gradient of cyclohexane/EtOAc, with EtOAc from 20 to 50%), affording the para-phenolic azobenzene (492 mg, 1.07 mmol) in 66% yield. The resulting diazocompound (480 mg, 1.05 mmol) was then methylated in acetone (9.6 ml) with

$K_2CO_3$ (431 mg, 3.12 mmol) and MeI (0,13 ml, 2.12 mmol) at room temperature for 16 h. The crude was concentrated and purified by column chromatography (cyclohexane/EtOAc, with EtOAc from 10 to 50%) affording the methylated diazocompound (491 mg, 1,04 mmol) in 99% yield. The resulting compound (415 mg, 0.88 mmol) was then treated in basic media (KOH in MeOH, reflux for 1 h) to remove the tosyl group. The reaction mixture was concentrated, dissolved in EtOAc and aq $KH_2PO_4$ (10%), the organic phases extracted with EtOAc, combined, washed with water and brine, and dried on $MgSO_4$. Purification by column chromatography (cyclohexane/EtOAc, with EtOAc from 20 to 50%) delivered azo-CA4 (220 mg, 0.69 mmol) with a 78% yield. Comparison to published NMR spectra[3] confirmed the successful synthesis (1H NMR (400 MHz, DMSO-d6) δ 9.47 (s, 1H), 7.46 (dd, J = 8.5, 2.4 Hz, 1H), 7.32 (d, J = 2.4 Hz, 1H), 7.18 (s, 2H), 7.12 (d, J = 8.6 Hz, 1H), 3.88 (s, 6H), 3.87 (s, 3H), 3.75 (s, 3H)).

### Sample preparation for serial crystallography
Crystal pellets were resuspended and the contents of two 0.6-ml tubes were combined and collected into 2-ml tubes. After centrifugation in a tabletop centrifuge (1000×g, 15 seconds), a concentrated crystal pellet of 45–60 µl volume was obtained. The pellets were extracted, transferred into PCR tubes, and mixed with the *trans*-azo-CA4 to a final concentration of 1 mM.

To switch the compound into the binding *cis* conformation, the tubes were then illuminated for at least 4 h at 385 nm before incorporation into hydroxyethylcellulose hydrogel (Sigma-Aldrich)[46]. Hydrogels containing 22% (w/v) hydroxyethylcellulose were prepared in syringes and left to cure for 5 days. Illuminated crystals were transferred into a Hamilton syringe and mixed into hydrogel matrix at a ratio of 4:7 using a three-way-coupler[47] before loading into the high-viscosity injector.

### Experimental setup and XFEL data collection
Time-resolved serial crystallographic data of light-induced azo-CA4 from tubulin were collected within 2 days of beamtime in September 2020 at the Alvra experimental station of SwissFEL. The X-ray source provided pulses with a photon energy of 12.1 keV and pulse energy between 300 and 550 µJ at a repetition rate of 100 Hz. To reduce X-ray scattering (background level), the sample chamber was pumped down to a pressure of 150 mbar and then refilled with helium to 500 mbar. This process was continuously repeated during the measurement. Data were collected using a Jungfrau 16M operated in 4M mode. Hit rate determination and initial data processing during the experiment were done with the online data analysis pipeline described in ref. [48].

TD1 crystals homogeneously incorporated into hydroxyethylcellulose were loaded into a high-viscosity injector connected to an HPLC pump[49]. The system extruded the crystal-loaded hydrogel matrix into the pump–probe interaction point through a 75 µm capillary at a flow rate of 5.3 µl per minute. In the interaction point, the probing X-ray pulses intersected with a circularly polarized pump beam originating from an optical parametric amplifier producing laser pulses with 350 fs duration $(1/e^2)$, 475 nm wavelength, and 10 µJ total energy in a focal spot of 65 µm $(1/e^2)$ diameter; corresponding to a maximal laser fluence of 301 mJ/cm² and laser power density of 861 GW/cm². In order to follow the reaction of light-induced azo-CA4 ligand release from tubulin between the pump laser and the probing, XFEL pulses were chosen at Δt = 1 ns, 10 ns, 100 ns, 1 µs, 10 µs, 100 µs, 1 ms, and 10 ms.

### Serial synchrotron crystallography
Serial synchrotron data were collected using a previously described setup[25]. Crystals embedded into hydroxyethylcellulose were injected in the intersecting X-ray and laser beam paths by a high-viscosity injector[49] with a 75-µm diameter nozzle.

The jet speed was set to 0.5 µm/ms and data frames were recorded using an Eiger16M detector with a frame rate of 100 Hz. The beam size was set to 20 × 5 µm² to cover the central part of the hydroxyethylcellulose column on one axis and extrusion by one frame on the other. In this way, we ensured the majority of crystals were exposed while minimizing similarities in consecutive frames for more reliable statistics. A photon flux of around $1.5 \times 10^{12}$ photons/s resulted in a radiation dose of about 5 kGy per recorded detector frame when calculated with RADDOSE-3D[50]. Depending on their size and orientation while traversing the beam this corresponds to between 5 and maximally 100 kGy per crystal. Illumination was done using light at 445 nm from a laser diode with a peak optical power of 5 mW (Roithner Lasertechnik). Light intensity at the interaction region was approximately 2.5 mW in a spot size of about 100 × 50 mm². The illumination time was set to approximately 100 ms by adjusting the distance between the laser and X-ray interaction regions.

### Data processing
Indexing, integrating, and merging of obtained data was performed using Crystfel version 0.9.1[51,52]. In detail, the peakfinder9 algorithm was used for peak detection, XGANDALF[53] was used to index obtained data; and peaks were integrated using -ring-radius=2,3,6; partialator options -model=unity, -iterations=1.0, -push-res=1.5 were used to merge selected patterns. A general resolution cutoff of 1.7 Å was chosen (see Supplementary Table 1) and refinements for collected dark-state and apo-state data were carried out to the full resolution range. Structural refinements of time-resolved data were carried out to 2.2 Å resolution due to data degradation when extrapolating data (see Supplementary Table 1). Serial synchrotron crystallography data obtained at the SLS showed stronger anisotropy than the data obtained at the SwissFEL. The data were therefore corrected for anisotropy using the STAR-ANISO server[54].

### Difference density maps calculation
Calculations of $F_{obs}^{light} - F_{obs}^{dark}$ difference maps were performed using PHENIX v1.19.2[55]. Briefly, the multi-scaling option was used excluding amplitudes smaller than 2 σ and exclusively considering the resolution range between 6 and 1.7 Å. For all calculated $F_{obs}^{light} - F_{obs}^{dark}$ difference maps, the phases of the refined dark state were used. $F_{calc}^{light} - F_{calc}^{dark}$ difference maps were computed to the same resolution using the CCP4 suite v8.0[56].

### Pearson correlation of difference maps
To compute Pearson correlations independent of individual dataset multiplicities, we reduced the number of patterns to match the time delay where we collected the least data (55,065 patterns) and calculated difference maps as stated above. These difference maps were then used to calculate the Pearson correlation of integrated difference electron densities[27] around all protein atoms and around the ligand and all residues lying within a 3-Å sphere around the ligand. Water molecules and hydrogen atoms were excluded from the selection. The density maps were integrated within a 2-Å radius around the atoms and densities below 1.5 σ were ignored. The resulting Pearson correlation values were normalized to the largest correlation, displayed on a gray scale, and values smaller than 0.3 were omitted from the figures for clarity.

### Integration of negative density around the ligand
To plot the release of azo-CA4 over time, we integrated the negative density around the ligand in the same way as the Pearson correlation data for the whole protein (1.5-Å radius, 3.0 sigma cutoff). The integrated negative density was normalized: the highest value (apo) was set to 1 the lowest negative ligand density (1 ms) was set to 0. Since apo data and time-resolved data from the XFEL experiment are not fully comparable due to a roughly fourfold difference in occupancy. The

negative density of the apo-state was downscaled to lower occupancy by assuming a roughly fourfold occupancy increase is similar to a fourfold increase in the number of observations, resulting in an estimated scaling factor of 0.5 (assuming signal to noise improves by $\sqrt{2}$ for doubling the number of observations).

### Data extrapolation

Calculation of extrapolated data was performed using the method described in ref. [29]. Briefly, the linear approximation was used: $F_{extra} = [100/A \times (F_{obs}^{light} - F_{obs}^{dark}] + F_{calc}$, with A representing the activation level in percent, $F_{calc}$ representing amplitudes of the high-resolution dark-state model and $F_{extra}$ representing the extrapolated structure factor amplitude. The activation level A, defined as the percentage of dark-state molecules being activated after pump laser pulse exposure, was determined as described in ref. [29]. Extrapolated data were calculated using a decreasing activation level A. Then the negative electron density in $2F_{extra} - F_{calc}$ maps calculated using the dark model phases was integrated around nitrogen atoms of the ligand which have strong negative difference density peaks on them in each of the light-activated structures and hence leave their position. The negative density was then plotted against the activation level. The intersection of the linear parts of the resulting graph (negative density is zero for not enough dark state subtracted and starts to linearly increase as dark is over-subtracted) was taken as the activation level. The activation levels determined in this way fluctuated between 18 and 27% but since all data were measured under the same conditions, we chose the average of 22% to refine the structures. As a further control, we determined the activation level using the re-appearance of dark state features in $2F_{extra} - F_{calc}$ maps [57,58], which confirmed our initial assignment.

### Structure determination and refinement

The structure of the dark state was solved by molecular replacement with 5NQT as a search model. Structural refinements of the model were done using PHENIX v1.19.2[55] with iterative cycles of manual adjustments made in Coot v0.9.8.1[59]. As a final step, the ligand was refined using Buster v2.10.4[60], employing the -qm option to use a quantum mechanics target for the ligand rather than restraints which may be a source of errors, especially for molecules that are not at their energy minimum. The structure of the apo-state was directly refined using the dark-state structure as a starting point.

Before the actual refinement and model building of the obtained time-resolved data, the negative amplitudes resulting from the extrapolation procedure were removed. The models were manually adjusted to best match observed difference map features, to produce similar $F_{calc}^{light} - F_{calc}^{dark}$ maps, and to fit extrapolated maps using the final dark model as a starting point. To match the observed difference density features using $F_{calc}(1\,ms) - F_{calc}^{dark}$ maps we chose to exclude Leu246 from XYZ refinement after manual adjustment for the 1 ms structure. Given the activation level of 22%, we choose to refine the predominant structural species by comparing the models to previous and consecutive structures.

### Binding pocket and protein backbone analysis

Protein interactions in the binding pocket were identified with the protein–ligand interaction profiler[61]. Changes in the volume of the colchicine site over time were determined using PyVOL[62] with the parameters: β-tubulin, minimum radius = 1.65; maximum radius = 3.8; minimum volume = 50, standard partitioning.

To visualize global protein backbone movements, alternative conformers were removed and structures truncated to poly-Ala using Phenix PDB tools[55]. Respective structural displacements of secondary structural elements were indicated using the PyMOL Python script Modevectors (by Sean M. Law, University of Michigan). Displacements greater than 0.5 Å were classified as significant (cutoff = 0.5).

For visualization purposes, only the displacement vectors of every second (skip = 2) Cα-backbone atom (atom = CA) were shown and the vectors were amplified by a factor of 6 (factor = 6). Additional settings involve the vector representation (head = 2.0, tail = 0.85, head_length = 3.0, cut = 0).

To find possible release pathways during the reorganization of the binding pocket, we used the program Caver 3.03[63] using the following settings: minimum probe radius = 1.9 Å, shell depth = 4 Å, and shell radius = 5 Å, azo-CA4 position as starting coordinate.

### Time-resolved spectroscopy measurements

The time-resolved transient absorption data were recorded using a home-built pump–probe setup, as described in detail in ref. [64]. In short, the fundamental laser pulses (1 mJ, 775 nm, 130 fs, 1 kHz) were provided by a Ti:Sa amplifier (Clark, MXR-CPA-iSeries). The pump pulses at 438 nm were generated using a home-built two-stage NOPA (noncollinear optical parametric amplifier) and sum frequency mixing. The pulses were pre-compressed in a prism compressor located between the two NOPA stages. White light continuum pulses (300–750 nm) were generated by focusing a fraction of the laser fundamental beam into a $CaF_2$-crystal (5 mm). The continuum pulses were split into the probe and reference beams. The reference beam was guided directly into a spectrograph, while the probe beam was focused at the sample position, then collected and directed into a second spectrograph. The spectrographs (AMKO Multimode) contained gratings with 600 grooves/mm blazed at 300 nm and a photodiode array. The instrument response function (IRF) of ~100 fs in the experiments was estimated from the pump–probe cross-correlation. Anisotropic contributions to the measurements were avoided by performing the experiments at the magic angle condition (54.7° pump–probe polarization difference). The samples were held in a fused silica cuvette with an optical path length of 1 mm which was constantly moved in the plane perpendicular to the direction of the probe pulse propagation to avoid the accumulation of photoproducts. To keep the azo-CA4 samples (bound and free) in the cis isomer state, the cuvette was continuously illuminated with a high-power LED at 385 nm. Global analysis of the experimental data was performed using OPTIMUS (www.optimusfit.org)[64] to recover the lifetimes describing the cis → trans photoisomerization kinetics. With the help of target analysis, a sequential kinetic scheme was fitted to the experimental data to obtain the times of the highest population of each kinetic state, which was helpful in determining the optimal time points for the XFEL experiments.

### Computational methods

**System preparation.** The starting models for simulations were built from the respective XFEL-based time-resolved structures. We removed the DARPin protein and reconstructed the βM-loop (res 272-288) using the one in chain D of PDB 4I4T as a template by employing the Prime[65] module of the Schrödinger 2020-4 suite. Analogously, $Mg^{2+}$ ion in chain B (β-tubulin) was added by superimposition with the one of chain D of 4I4T structure. Calcium ions were removed from the model.

The final model consisted of an αβ-tubulin heterodimer in complex with azo-CA4, GTP, and GDP molecules bound to the α- and β-tubulin subunits, respectively, as well as their associated $Mg^{2+}$ ions and their coordinating water molecules. The resulting protein structure possessed 437 out of 451 residues of α-tubulin (UniProtKB ID P81947) and 431 out of 445 residues of β-tubulin (UniProtKB ID Q6B856). Missing residues belonging to the intrinsically disordered C-terminal tails of α- and β-tubulin were not modeled and C-termini were capped with N-methyl amide (NME) groups. Residue protonation states were evaluated at pH 7.0 using the Protein Preparation Wizard tool[66] implemented in the Schrödinger 2020-4 suite[67]. The αβ-tubulin heterodimer structure was solvated with the TIP3P-model[68] for water molecules in a truncated octahedron box using 12 Å as the minimum

distance between the protein and the box edges. The system was neutralized by adding $Na^+$ ions resulting in a total of about 122k atoms. The atomistic force field Amber-ff14SB[69] was used for all simulations. Parameters for $Mg^{2+}$ ions and the GTP and GDP molecules were developed by Allner et al.[70] and Meagher et al.[71], respectively.

Concerning the ligand, partial atomic charges were obtained with the Restrained Electrostatic Potential (RESP) method on a DFT-optimized trans conformation at the B3LYP/6-31 g* level. The geometric optimization was carried out with Terachem v1.94. The General Amber Force Field (GAFF) was used, with custom parameters for the three main dihedral angles: 1-1a-1b-1', 6-1-1a-1b, 1a-1b-1'-6'.

To obtain such optimized parameters, we fitted the GAFF energy curves associated with a rigid torsion of the dihedral angles to the corresponding DFT energy curves. For each dihedral, we proceeded as follows: we started out with the geometrically optimized *trans* structure. We then performed a clockwise rigid rotation with 15° steps followed by an analogous counter-clockwise rigid rotation. For each rotation step we performed a DFT geometric optimization restraining the rotated dihedral, and we evaluated the energy. We retained the minimum energy among the clockwise and anticlockwise rotation cycles to overcome hysteresis. The three resulting DFT energy profiles consisting of 24 evaluation points were used as a target for the fitting procedure. DFT calculations were performed with Terachem v1.94, at the B3LYP/6-31 g* level. The fitting procedure was carried out separately for the three energy profiles with mdgx, part of AmberTools21. The αβ-tubulin heterodimer system was assembled with the LEaP tool implemented in the AmberTools21 software package[67].

The designed models for QM calculations consisted of protein residues α(99-101, 178-181), β(237-242, 248-259, 314-321, 349-354, 378-380), with the respective N- and C- termini capped with acetyl (ACE) and NME groups, the azo-CA4 molecule, and its two closest water molecules. They resulted in 743 atoms.

**Molecular dynamics simulations.** Plain molecular dynamics simulations starting from the tubulin-trans-azo-CA4-tubulin complex 1 ms structure were performed using the Amber20 engine[67]. The system, after a preliminary stage of energy minimization (1000 steps of steepest descent and 9000 of conjugate gradient), was heated to the target temperature of 300 K with a 250 ps ramp using an integration time-step of 1 ps and the weak-coupling algorithm, followed by another 500 ps of simulation in the NVT ensemble after switching to an integration time-step of 2 ps. Subsequently, in order to equilibrate box dimensions, the system was simulated for 1 ns in the NPT ensemble using the Berendsen barostat. For these initial stages, heavy atoms were harmonically restrained with a force constant of 5 kcal/mol/Å². Finally, a last stage of 5 ns in the NVT ensemble, using the Langevin thermostat with a collision frequency g of 2.0 ps⁻¹. Bonds involving hydrogen atoms were restrained, and a short-range, non-bonded cut-off of 8 Å was applied, whereas long-range electrostatics were treated with the particle mesh Ewald (PME) method[72]. Periodic boundary conditions (PBC) were applied. After the equilibration stage, a cumulative 30-µs-long MD simulation was conducted with an integration time-step of 2 fs in the NVT ensemble at a target temperature of 300 K.

In order to evaluate the $\Delta\Delta G_{cis\text{-}trans}$, well-tempered metadynamics (WTMD) simulations[73] were performed with GROMACS 2020.2[74] and PLUMED 2.6.1[75]. We performed WTMD enhancing the sampling of the two dihedral angles 1-1a-1b-1' and 6-1-1a-1b for both the bound and the unbound state. The bias factor, the deposition rate, the initial hill height, and the hill width were set to 45, 0.5 ps⁻¹, 2.5 kJ·mol⁻¹, and 0.3 rad, respectively.

The WTMD simulation of the bound state was carefully designed to sample conformations compatible with the XFEL-based time-resolved structures. A positional harmonic restraint with K = 100 kJ/mol nm² was introduced on the backbone of the protein together with the supervision of the hydrogen bond to the

backbone carbonyl of αThr179. With this regard, the simulation was stopped and restarted with regenerated velocities whenever the H-bond was broken. The resulting trajectory of 400 ns was reweighted in order to reconstruct the free energy profile. Block analysis with a block size of 50 ns was performed to assess convergence and to plot 95% confidence interval error bands. We performed WTMD enhancing the sampling of the two dihedral angles 1-1a-1b-1' and 6-1-1a-1b. The bias factor, the deposition rate, the initial hill height, and the hill width were set to 45, 0.5 ps⁻¹, 2.5 kJ mol⁻¹ 740, and 0.3 rad, respectively.

The WTMD simulation of the unbound state was performed with the same parameters used for the bound state. In this case, we simulated a single ligand molecule in a solvated cubic box using 10 Å as the minimum distance between the ligand and the box edges. After standard minimization, thermalization and equilibration phases WTMD was carried out for 400 ns. The resulting trajectory was reweighted in order to reconstruct the free energy profile. Block analysis with a block size of 50 ns was performed to assess convergence and to plot 95% confidence interval error bands.

### Reporting summary
Further information on research design is available in the Nature Portfolio Reporting Summary linked to this article.

## Data availability
The PDB entry 4I4T (tubulin-RB-TTL complex) was used as starting point to prepare the QM simulations. For molecular replacement, we used PDB entry 5NQT (tubulin-darpin complex). We used the PDB entry 5SYE (taxol-stabilized microtubule) for the tubulin conformation found in microtubules. Coordinates and structure factors of the presented serial femtosecond crystallographic structures have been deposited in the PDB database under accession codes 7YYQ (dark) and 7YZ3 (apo). The serial synchrotron data have been deposited under 7YZ6 (dark) and 7YZ5 (100 ms). Models refined against extrapolated data, extrapolated structure factors, and light data used for data extrapolation have been deposited under accession codes 7YYV (1 ns), 7YYW (10 ns), 7YYX (100 ns), 7YYY (1 µs), 7YYZ (10 µs), 7YZ0 (100 µs), 7YZ1 (1 ms) and 7YZ2 (10 ms). The start and end coordinates of the molecular dynamics simulations are provided in Supplementary Data 1 and 2.

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

## Acknowledgements

We are grateful for the excellent support from the PSI Crystallization Facility and the Macromolecular Crystallography group during the growing and testing of crystals at the Swiss Light Source. The Biomolecular Structure and Mechanism Program of the Life Science Zürich Graduate School is acknowledged for its academic framework for our graduate students. This project was founded by the following agencies: The Swiss National Science Foundation project grants 31003A_179351 (to J.S.), to 310030_197674 (to T.W.), 310030_192566 (to M.O.S.) and the NCCR:MUST (to C.M. and J.S.), the Swiss Nanoscience Institute SNI #1904 (to M.C.), Deutsche Forschungsgemeinschaft, Project WA 1850/4-3 (C.S. and J.W.).

## Author contributions

The project was coordinated and led by J.S. following input on time-resolved serial crystallographic data acquisition and evaluation from T.W., computer simulations from A.C., and time-resolved spectroscopy from J.W. and tubulin structural biology from M.O.S. The TD1 protein complex was purified and crystallized by M.Wr. with help from A.F., N.G., and other members of the team during the large-scale preparation phase. The azo-CA4 compound was synthesized by T.Ma. Crystal injection was optimized by M.Wr., A.F., and D.J. The lipidic cubic phase injector was operated and aligned during the beamtime by A.F. D.Ga., and D.J. The endstation including the laser system was aligned and operated by P.J.M.J., D.O., K.N., C.C., J.B., C.B., and C.M. Sample preparation during the beamtime was done by M.Wr., A.F., M.C., R.S., D.K., and H.G. Data processing during the beamtime was initially done using the online data analysis pipeline established by K.N. and D.O. followed by manual optimization by S.B. and P.S. with instructions from T.W. The synchrotron setup was aligned and optimized by F.D., D.J., and M.Wa. Final structures were refined by M.Wr. and T.W. and interpreted together with J.S. Quantum chemical calculations were done by M.F. and D.Gi. under supervision of A.C. The time-resolved spectroscopic experiments were done by N.G., T.W., and C.S. and interpreted together with M.B. and J.W. The manuscript was written by T.W., M.Wr., M.O.S., and J.S. with further suggestions from the other authors.

## Competing interests

The authors declare no competing interests.
