## [Peer Review File · Nature Communications]

Watching the release of a photopharmacological drug from tubulin using time-resolved serial crystallographyReviewer #1 (Remarks to the Author):

This is a very good paper, and I highly recommend publication. It uses upcoming capabilities of X-ray free electron lasers to determine a time-resolved series of structures of photoactive proteins. What is new, to the best of my knowledge, that this is an artificially photoswitchable system, whereas previous examples studied natural photoactive proteins (e.g. bacteriorhodopsin). What is also new is the fact that the photoswitch is on the surface of the protein, and is not covalently bound, while the photoactive chromophore in natural systems is embedded in the protein. While I am not an expert in crystallization, I could imagine that this poses problems during crystallisation; for example, it is the cis form which binds, which however is thermodynamically less stable. The information is given (hidden) in Methods, but I feel the authors could advertise the novelty of this work, in comparison to bacteriorhodopsin and the like, a bit more. I have always been under the impression that time-resolved crystallography is not possible for protein systems of this sort.

From a science point of view, the paper studies ligand unbinding, which obviously is a very important process for all sorts of protein-drug interactions. Time-resolved crystallography can do so in unprecedented microscopic detail, both concerning the aspect of time as well as structure, and this paper very nicely demonstrates that.

To make the connection to the "normal" (biochemistry) approach of ligand binding, I would like to ask the authors to add a bit of a discussion along the following lines:

An unbinding time (k_{off}) of 1 ms is pretty fast, and in essence it says that the binding affinity in trans is very low. Quite a few examples exist in literature where people explored the binding affinity of photoswitchable ligands, and found almost always that the change in binding affinity between the two states is relatively modest, i.e., K_d changing by factors of 2-10. Here it is said that it is changing by a factor 4300, which is exceptionally large, but that number is obtained from computation only (as I understand it). There must be experimental data on the binding affinities in solution in literature (in Ref. 3?). If not, the authors must measure those numbers, since they set really the stage for these experiments. If the binding affinities are known, one can also estimate k_{off} , since k_{on} is governed by diffusion and a number that doesn't vary a lot. Or, measure k_{on}/k_{off} in solution, which then can be compared to the 1ms reported here.

This brings me to the real question: To what extent can ligand unbinding in a crystal be compared to that in solution. Many questions occurred to me and there really should be a discussion on that: Where does the ligand go? Can it be seen in X-ray, or is it diffuse? How much water is there in the crystal, and what is the effective concentration of protein in the crystal? I guess very large, so even if the binding affinity in trans is, say, $K_d > 1\text{mM}$ (which would effectively be considered no binding anymore), due to the even higher protein concentration in the crystal, it should still bind. Also, these are microcrystals. What is the ratio of proteins at the surface of these microcrystals, relative to those in the interior? The former would be directly exposed to the solvent, and my argument above would not apply anymore. I understand the ligand is added to the solution only after crystallisation. Can they diffuse into the crystal? Would all proteins have a ligand bound, or only those at the surface? Some of this information might be hidden in Method, but to appreciate this paper, it would be good to have it in the main paper, as it is more than just a technical detail.

Overall, I congratulate the authors for this beautiful work.

Reviewer #2 (Remarks to the Author):

Dear Editor,

This work addresses the light-triggered mechanism of release of an azobenzene

derivative of combretastatin from the colchicine binding site at the intra-dimeric interface of a tubulin heterodimer. In opinion of this reviewer, the outcomes of this manuscript will be of significance for the design of novel phototherapeutics targeting the tubulin protein.

Concerning MD simulations, the study is solid, thoroughly designed and executed. Also, the data is well presented in the paper. Minor comments/questions in this regard were raised during the revision process, as detailed below:

1. Reweighted free energy profiles for the interconversion between cis-trans isomers in free/bound states were not provided. This information could be relevant to compare the results with the previous findings obtained for cis/trans combretastatin (reference 30).

2. Information regarding the collective variables used to perform WTM calculations was not provided. This information is required for the work to ensure reproducibility.

3. How many independent WTM simulations were carried out? Did independent simulations converge to similar values for the free energy difference between the cis/trans isomers in the free and bound states? Do the authors have an estimate of the error involved in calculating the Delta-Delta binding free energies between cis/trans isomers?

4. As indicated on page 33, plain MD simulations starting from the trans isomer were carried out. Did authors explore beginning with the cis isomer to then conduct WTM simulations?

5. Did the authors check the mobility of the betaT7 loop in ligand-free MD simulations of the free energy required to induce the conformational change in the protein such that the colchicine site is suitable for ligands' binding (as a blank system)?

Other aspects:

Page 2, line 55. References 3 and 4 appear to be misplaced.

Page 27, line 729: The ratio of trans/cis isomers of compound 6 is missing.

Reviewer #3 (Remarks to the Author):

In conventional structure-based drug design (SBDD), a crystal structure without information on protein/enzyme motion has been used as a template to design compounds that bind to the protein/enzyme. This is because conventional synchrotron radiation crystallography can only obtain a snapshot of the crystal structure of a protein molecule in a state of arrested motion. However, the actual interaction process between protein and ligand (substrate/product) involves more or less structural changes in the ligand binding site of the protein. So, for example, by obtaining snapshots of an enzyme-substrate analog complex, an enzyme-reaction intermediate analog complex, and an enzyme-product complex, in which the reactions do not proceed, it has been attempted to imagine and draw a whole picture of the enzyme reaction cycle from those three motionless structures.

Recent developments of time-resolved serial crystallography technology have changed this situation. It is now possible to capture snapshots of molecules in motion in crystals and visualize the structure of reactions in progress, which had previously been a black box. This means that a new era of drug design will be pioneered, focusing on the motions of live proteins, which was not possible with the conventional SBDD.

The present work by Wranik et al. is not only important as a new attempt toward above goal, but it is also significant because it provides insights that may lead to a general

understanding of protein-ligand interactions. The authors accomplished this by analyzing the interaction of the chemically synthesized light-regulable molecular tool azo-CA4 with tubulin using time-resolved serial crystallography and visualizing molecular motions at near atomic resolutions in the nanosecond to millisecond range, which is much shorter than the time resolution limit observable with cryo-EM and other techniques. It also seemed reasonable that the authors complemented the findings from crystallographic data with molecular dynamics simulations, but the details of the computational methods are difficult to evaluate from my expertise.

I would like to request that the authors answer the following questions and comments.

About ligand concentrations:

Line 617 describes that the authors evaluated using 0.5 mM cis-azo-CA4 or 5 mM cis-azo-CA4/trans-azo-CA4, while Line 627 describes that they tested with 1.25 mM cis-azo-CA4 and Line 735 with 1 mM trans-azo CA4.

Why was the sufficiently high 5 mM ligand concentration not employed in the time-resolved experiments, but a lower concentration?

Figure 1D:

Although there are only three Figure panels, the time arrow covers from 1ns to 100 ms, making it difficult to tell which time point structure is indicated. The explanation that the center panel corresponds to 1 ms is particularly confusing because it is misaligned with the arrow. It would be better to provide a more detailed explanation or modify the drawing to make it easier to understand.

Figure 3A:

If possible, dissociation constants (Kd values) should also be listed.

Supplementary Table 1 and 2:

Table titles should be written above the Table.

Supplementary Table 2:

The authors should comment on why the Rwork and Rfree values of the pump-probed datasets are significantly higher (worse) than those of the other datasets. Did the pump light exposure itself damage the crystals?

Line 126 :

The sentence "here we focus on the biologically most relevant temporal domain starting from the nanoseconds." may need a reference.

Supplementary Figure 4, Line 648:

"Positive and negative difference density are displayed in red and green, respectively." Is the description of the color for the density correct?

Line 731:

"Sample preparation serial crystallography"  "Sample preparation for serial crystallography" ?

REPLY TO REVIEWER COMMENTS

Reviewer #1 (Remarks to the Author):

- This is a very good paper, and I highly recommend publication. It uses upcoming capabilities of X-ray free electron lasers to determine a time-resolved series of structures of photoactive proteins. What is new, to the best of my knowledge, that this is an artificially photoswitchable system, whereas previous examples studied natural photoactive proteins (e.g. bacteriorhodopsin).

Our thanks for the positive evaluation and interest in our work. The insightful have been integrated as follows to further improve our manuscript.

- What is also new is the fact that the photoswitch is on the surface of the protein, and is not covalently bound, while the photoactive chromophore in natural systems is embedded in the protein. While I am not an expert in crystallization, I could imagine that this poses problems during crystallisation; for example, it is the cis form which binds, which however is thermodynamically less stable. The information is given (hidden) in Methods, but I feel the authors could advertise the novelty of this work, in comparison to bacteriorhodopsin and the like, a bit more. I have always been under the impression that time-resolved crystallography is not possible for protein systems of this sort.

Yes, the referee is right, the use of a photochemical affinity switch for time-resolved crystallography is a novel approach and we stress its potential throughout the main text and supplementary figures 1-3. Depending on the system, it would be possible to use either cis or trans but in our case only the cis compound is binding. As we describe in the method section, we generated the cis isomer by illumination just before loading the injector. The thermal relaxation of the cis isomer is so slow that it does not interfere with the experiment (also compare ligand characterization in Supplementary Figure 3). To better highlight the overall novelty of our approach to adapt ligands developed for photopharmacology for time-resolved crystallography we have updated the corresponding section

*“... and the technology is now available at a growing number of synchrotrons and X-ray free-electron lasers (XFELs)^{17,18}. With increased availability of beamtime at these advanced X-ray sources, future challenges will shift towards enabling dynamic studies on a wide range of relevant biological targets. Photopharmacology could be an important part of the solution as the field already developed a variety of photosensitive compounds that can be utilized as efficient triggers to activate diverse soluble and membrane protein of medical interest (for some examples, see **Supplementary Figure 2**).”*

- From a science point of view, the paper studies ligand unbinding, which obviously is a very important process for all sorts of protein-drug interactions. Time-resolved crystallography can do so in unprecedented microscopic detail, both concerning the aspect of time as well as structure, and this paper very nicely demonstrates that.

We fully agree with the reviewer and hope that he/she agrees with our choice to focus on the biological impact in the main text, while exploring methodological aspects in the supplementary and methods section. Generally, we see both as important and interesting but had to split them for better accessibility of the manuscript.

- To make the connection to the “normal” (biochemistry) approach of ligand binding, I would like to ask the authors to add a bit of a discussion along the following lines: An unbinding time (k_{off}) of 1 ms is pretty fast, and in essence it says that the binding affinity in trans is very low. Quite a few examples exist in literature where people explored the binding affinity of photoswitchable ligands, and found almost always that the change in binding affinity between the two states is relatively modest, i.e., K_d changing by factors of 2-10. Here it is said that it is changing by a factor 4300, which is exceptionally large, but that number is obtained from computation only (as I understand it). There must be experimental data on the binding affinities in solution in literature (in Ref. 3?). If not, the authors must measure those numbers, since they set really the stage for these experiments. If the binding affinities are known, one can also estimate k_{off} , since k_{on} is governed by diffusion and a number that doesn't vary a lot. Or, measure k_{on}/k_{off} in solution, which then can be compared to the 1 ms reported here.

Our results show the unbinding process to take place somewhere between 1 ms (bound) and 10 ms (predominantly unbound), which indeed is short and indicates a low affinity of trans-azo-CA4 as the reviewer points out. In the literature, the light-dependent potency difference between the cis- and the trans-state is described to be 250 in cellular assays [Shiva et al., EJMC, 2018, Borowiak et al., Cell, 2015]. Radioligand scintillation proximity assays have been used to measure binding to tubulin in solution and indicate a EC_{50} of around 30 μM for illuminated azo-CA4 (in comparison to 0.16 μM for the parent compound CA4) but no significant competitive binding for the relaxed trans-azo-CA4 [Borowiak et al., Cell, 2015]. The potency difference between the light-on and light-off states of a compound is one of the most important parameters photopharmacologists want to optimize. In our manuscript, we use a combination of simulation and time-resolved crystallography to reveal the underlying mechanisms on this prominent example for a photochemical affinity switch. Admittedly, the decrease in the affinity of trans-azo-CA4 obtained by this analysis is unusually large but overall fits well to the available ligand binding data. To better clarify this aspect, we have changed affinity to the clearer potency in Figure 2 and have added the following to the revised manuscript.

“This calculated change is in good agreement with radioligand competition assays that indicated an EC_{50} value of $\sim 30 \mu M$ for illuminated azo-CA4 (representing a mixture of cis-trans isomers) but no significant competitive binding of the relaxed trans-azo-CA4 to tubulin {Borowiak, 2015 #190}.”

- This brings me to the real question: To what extent can ligand unbinding in a crystal be compared to that in solution. Many questions occurred to me and there really should be a discussion on that: Where does the ligand go? Can it be seen in X-ray, or is it diffuse? How much water is there in the crystal, and what is the effective concentration of protein in the crystal? I guess very large, so even if the binding affinity in trans is, say, $K_d > 1 \text{ mM}$ (which would effectively be considered no binding anymore), due to the even higher protein concentration in the crystal, it should still bind. Also, these are microcrystals. What is the ratio of proteins at the surface of these microcrystals, relative to those in the interior? The former would be directly exposed to the solvent, and my argument above would not apply anymore. I understand the ligand is added to the solution only after crystallisation. Can they diffuse into the crystal? Would all proteins have a ligand bound, or only those at the surface? Some of this information might be hidden in Method, but to appreciate this paper, it would be good to have it in the main paper, as it is more than just a technical detail.

It is an old obstacle of structure biology to determine to what extent the structure and function of a protein within a crystal corresponds to those of the same protein in its native environment. As shown in Supplementary Figure 3D, we used time-resolved UV/Vis spectroscopy to determine the difference in ligand isomerization when in solution and bound to the protein up to 1 ns after photoactivation. In this temporal regime, the recorded structural snapshots are in good agreement with the spectroscopic data. Details of this analysis will be discussed in a separate publication focusing on the ultrafast isomerization and relaxation of the ligand.

For longer time delays, we would like to refer to MD simulations describing the release of trans-Combretastatin A4 to the solvent after structural displacements of the betaT7-loop [Gaspari et al., Chem, 2017]. In our structural study, we observed adaptations of the betaT7-loop residues over the 1us – 1ms time range that opened up a possible release pathway at 1ms. The ligand diffuses out of the binding pocket between 1ms and 10ms, which is in excellent agreement with binding studies and simulations as discussed above.

The diffusion process itself cannot be followed by X-ray crystallography as the method relies on ordered states within the crystal lattice. However, the MD simulations shown in Figure 3D illustrate the centers of masses of the ligand when diffusing out of its protein binding pocket towards the solvent environment. The solvent content of our tubulin/D1 crystals is 45.5% and roughly in the middle of the range of 25-90% reported for protein crystals [Weichenberger et al., Act. Crystallograph. D, 2015]. This makes it very easy for the ligands to diffuse in and out of the crystal, as is common practice in crystal soaking experiments for structure-guided drug design projects. Ligand binding is therefore not limited to the protein molecules exposed to the surrounding solvent at the crystal surface. At the start of the reaction, we observed full occupancy within the binding pocket, which drops later when the ligand is released in the milliseconds time regime (Figure 1C). Data and rationale for choosing azo-CA4 concentrations is presented in Supplementary Figure 3. For an excellent recent review of ligand diffusion within crystals, we would like to refer the reviewer to Schmidt, Crystals, 2020, which is now also cited in the conclusion section of our revised manuscript.

“It is furthermore difficult to predict how ligands diffuse in a crystal, as this depends on crystal shape, size, packing and solvent content among other external parameters {Schmidt, 2020 #427}. Starting a time-resolved experiment with a prebound ligand is an elegant solution to the problem, even though it would be interesting to study to what extent ligand binding and unbinding follow the same principles.”

- Overall, I congratulate the authors for this beautiful work.

Again, many thanks for the interesting and helpful comments further opening the manuscript to a broader audience.

Reviewer #2 (Remarks to the Author):

- Dear Editor,
This work addresses the light-triggered mechanism of release of an azobenzene derivative of combretastatin from the colchicine binding site at the intra-dimeric interface of a tubulin heterodimer. In opinion of this reviewer, the outcomes of this manuscript will be of significance for the design of novel phototherapeutics targeting the tubulin protein.
Concerning MD simulations, the study is solid, thoroughly designed and executed. Also, the data is well presented in the paper. Minor comments/questions in this regard were raised during the revision process, as detailed below:

We thank the reviewer for the positive assessment and have integrated the suggestions as detailed below.

- 1. Reweighted free energy profiles for the interconversion between cis-trans isomers in free/bound states were not provided. This information could be relevant to compare the results with the previous findings obtained for cis/trans combretastatin (reference 30).

We thank the reviewer for pointing this out. Free energy profiles of the interconversion between the two isomers, for both the free and the bound states, are now reported in Supplementary Figure 5 of our revised manuscript.

- 2. Information regarding the collective variables used to perform WTM calculations was not provided. This information is required for the work to ensure reproducibility.

We performed WTMD enhancing the sampling of the two dihedral angles 1-1a-1b-1' and 6-1-1a-1b. The bias factor, the deposition rate, the initial hill height, and the hill width were set to 45, 0.5 ps⁻¹, 2.5 kJ·mol⁻¹ 740, and 0.3 rad, respectively.

The WTMD simulation of the unbound state was performed with the same parameters used for the bound state. In this case, we simulated a single ligand molecule in a solvated cubic box using 10 Å as minimum distance between the ligand and the box edges. These data are now reported in the Materials and Methods section of the revised manuscript.

- 3. How many independent WTM simulations were carried out? Did independent simulations converge to similar values for the free energy difference between the cis/trans isomers in the free and bound states? Do the authors have an estimate of the error involved in calculating the Delta-Delta binding free energies between cis/trans isomers?

A single WTM simulation was carried out with a custom protocol to sample conformations compatible to the XFEL-based time-resolved structures. Such custom protocol consists of a positional harmonic restraint with $K = 100 \text{ kJ/mol nm}^2$ on the backbone of the protein together with the supervision of the hydrogen bond to the backbone carbonyl of a Thr179 (the simulation was stopped and restarted with regenerated velocities whenever the H-bond was broken). Block analysis with block size of 50 ns was performed to assess convergence. The reconstructed FES is now shown in Fig. S5 together with 95% confidence interval error bands.

- 4. As indicated on page 33, plain MD simulations starting from the trans isomer were carried out. Did authors explore beginning with the cis isomer to then conduct WTM simulations?

The MD was carried out with the trans isomer and the cis was simulated for some ns for system equilibration before the WTM studies. The choice of using only the trans isomer for plain MD simulations was driven by its much lower affinity to the binding site, in order to enhance the probability of observing the unbinding event.

- 5. Did the authors check the mobility of the betaT7 loop in ligand-free MD simulations of the free energy required to induce the conformational change in the protein such that the colchicine site is suitable for ligands' binding (as a blank system)?

No, we have not done ligand free simulations. Certainly, this may be an aspect to be further investigated in subsequent studies with tubulin and indeed in a previous study we found experimentally that the β -T7 loop is quite flexible in the apo-state and the colchicine bound state at room-temperature [Weinert et al, Nature Comm., 2017]. Such dynamics of the apo-protein will be most relevant when adopting our approach to differentiate between induced fit and conformational selection mechanisms in time-resolved binding experiment.

- Other aspects:
- Page 2, line 55. References 3 and 4 appear to be misplaced.

We agree that the references could have been better placed. We have moved Ref 3 to the first part of the sentence and replaced Ref 4 (referring to the first structure showing a ligand in the colchicine site) with a new reference to a structure with combretastatin A4.

The revised part now reads "A prominent example is an azobenzene derivative of combretastatin A4 (azo-CA4){Borowiak, 2015 #190}, a photopharmacological compound that binds the colchicine-site of the $\alpha\beta$ -tubulin heterodimer (hereafter called tubulin){Gaspari, 2017 #222}."

- Page 27, line 729: The ratio of trans/cis isomers of compound 6 is missing.

Line 729 refers to the activation level of 22 percent. At 1 ns this corresponds to 22 percent trans state.

After seeing the comment, we realized it might be confusing to switch from the cis/trans to the E/Z nomenclature in the section describing the synthesis of azo-CA4. We now use cis/trans throughout and also added a sentence that dark relaxation leads to the pure trans compound used in our experiments.

Reviewer #3 (Remarks to the Author):

- In conventional structure-based drug design (SBDD), a crystal structure without information on protein/enzyme motion has been used as a template to design compounds that bind to the protein/enzyme. This is because conventional synchrotron radiation crystallography can only obtain a snapshot of the crystal structure of a protein molecule in a state of arrested motion. However, the actual interaction process between protein and ligand (substrate/product) involves more or less structural changes in the ligand binding site of the protein. So, for example, by obtaining snapshots of an enzyme-substrate analog complex, an enzyme-reaction intermediate analog complex, and an enzyme-product complex, in which the reactions do not proceed, it has been attempted to imagine and draw a whole picture of the enzyme reaction cycle from those three motionless structures.

Recent developments of time-resolved serial crystallography technology have changed this situation. It is now possible to capture snapshots of molecules in motion in crystals and visualize the structure of reactions in progress, which had previously been a black box. This means that a new era of drug design will be pioneered, focusing on the motions of live proteins, which was not possible with the conventional SBDD.

The present work by Wranik et al. is not only important as a new attempt toward above goal, but it is also significant because it provides insights that may lead to a general understanding of protein-ligand interactions. The authors accomplished this by analyzing the interaction of the chemically synthesized light-regulable molecular tool azo-CA4 with tubulin using time-resolved serial crystallography and visualizing molecular motions at near atomic resolutions in the nanosecond to millisecond range, which is much shorter than the time resolution limit observable with cryo-EM and other techniques. It also seemed reasonable that the authors complemented the findings from crystallographic data with molecular dynamics simulations, but the details of the computational methods are difficult to evaluate from my expertise.

We fully agree with the evaluation and the relevance for structure-based drug design. It is our hope that this will be only the first in a long series of similar studies on other interesting targets.

- I would like to request that the authors answer the following questions and comments.

About ligand concentrations:

Line 617 describes that the authors evaluated using 0.5 mM cis-azo-CA4 or 5 mM cis-azo-CA4/trans-azo-CA4, while Line 627 describes that they tested with 1.25 mM cis-azo-CA4 and Line 735 with 1 mM trans-azo CA4.

Why was the sufficiently high 5 mM ligand concentration not employed in the time-resolved experiments, but a lower concentration?

In our time-resolved crystallographic experiments, we minimized the amount of free ligand to reduce the chance of rebinding events and to not unproductively absorb activating laser light. The process of this initial characterization is illustrated in Supplementary Figure 3 for all readers interested into the methodology or to transfer our approach to other targets.

- Figure 1D:
Although there are only three Figure panels, the time arrow covers from 1ns to 100 ms, making it difficult to tell which time point structure is indicated. The explanation that the center panel corresponds to 1 ms is particularly confusing because it is misaligned with the arrow. It would be better to provide a more detailed explanation or modify the drawing to make it easier to understand.

We wanted to show the 1 ms map because it is an important time delay where the binding pocket opens. But yes, it is somewhat confusing that it was not directly above the 1 ms mark of the time arrow in the figure. To fix the problem we have replaced the 1 ms electron density with that of the 100 us time delay. In addition, we have added markers indicating the time delays for each of the panels on the time arrow. Examples for difference electron densities obtained at all nine logarithmically spaced time delays are shown in the supplementary information.

- Figure 3A:
If possible, dissociation constants (Kd values) should also be listed.

Based on a similar request from Reviewer 1, we have added EC50 values from radioligand scintillation proximity assays to the main text.

- Supplementary Table 1 and 2:
Table titles should be written above the Table.

We have implemented the change as suggested.

- Supplementary Table 2:
The authors should comment on why the Rwork and Rfree values of the pump-probed datasets are significantly higher (worse) than those of the other datasets. Did the pump light exposure itself damage the crystals?

As we did not observe a reduction in hit rate or resolution between dark and light data from the same runs, the pump light did no damage diffraction quality of the crystals. The reduced Rwork and Rfree values rather originate from the extrapolation of different structure factor amplitudes used to refine the active states. This procedure is necessary since it only activates a fraction of molecules within a crystal (in our case an average of 22%). But unfortunately, it is known to introduce additional errors that worsens the refinement statistics. However, we would like to stress that the effect stays well within the limits of what has been observed in comparable time-resolved studies cited throughout the manuscript.

- Line 126 :
The sentence “here we focus on the biologically most relevant temporal domain starting from the nanoseconds.” may need a reference.

*In response we have changed the section to “We have characterized the ultrafast regime and initial photochemical reaction using transient absorption spectroscopy (**Supplementary Figure 3**); however, here we focus on the temporal domain starting from the nanoseconds where the biochemically most relevant conformational changes in the protein are to be expected {Orville, 2020 #365}.”*

- Supplementary Figure 4, Line 648:
"Positive and negative difference density are displayed in red and green, respectively."
Is the description of the color for the density correct?

Many thanks for spotting this mistake. We now follow the same density color code in all figures and have adjusted the figure legend accordingly.

- Line 731:
“Sample preparation serial crystallography”  “Sample preparation for serial crystallography” ?

We have introduced the suggestion of the reviewer.

Reviewer #1 (Remarks to the Author):

I would have hoped that the authors add a bit more to my last point ("To what extent can ligand unbinding in a crystal be compared to that in solution."). It is not about the question whether structure is the same in a crystal or in solution, but about the fact that ligand binding is a bimolecular process, that is concentration dependent, and the very concept of a concentration is difficult in a crystal. But again, it is excellent work, it is their decision, and I certainly don't want to hold up publication.